# Bayesian information theoretic model-averaging stochastic item selection for computer adaptive testing

## Abstract

Computer Adaptive Testing (CAT) aims to accurately estimate an individual's ability using only a subset of an Item Response Theory (IRT) instrument. A secondary goal is to ensure diverse item exposure across different testing sessions, preventing any single item from being over or underutilized. In CAT, items are selected sequentially based on a running estimate of a respondent's ability. Prior methods almost universally see item selection through an optimization lens, motivating greedy item selection procedures. While efficient, these methods tend to have poor item exposure. Existing stochastic methods for item selection are ad-hoc, where item sampling weights lack theoretical justification. In this manuscript, we formulate CAT as a Bayesian model averaging problem. At each step, we sample the next item in a manner where the Frequentist item sampling statistics correspond to Bayesian model averaging in the space of next-item ability estimates. This view of the CAT item selection problem also defines the natural criterion of the ability discrepancy: the KL divergence between the unknown next-item ability estimate and the unknown true full item bank ability estimate. We tested our new method on the eight independent IRT models that comprise the Work Disability Functional Assessment Battery, comparing it to prior art. We found that our stochastic methodology had superior item exposure while not compromising in terms of test accuracy and efficiency.

## 1 Introduction

The combination of Item Response Theory (IRT) and Computer Adaptive Testing (CAT) forms the dominant methodology backing the use of exams for ability assessment. High profile examples of this pairing include the Graduate Management Admission Test (GMAT) (Kingston et al., 1985; Rudner, 2010), the nursing National Council Licensure Examination (NCLEX) (Woo & Dragan, 2012), the National Registry of Emergency Medical Technicians (NREMT) (Ventura et al., 2021), and the Armed Services Vocational Aptitude Battery (ASVAB) (Segall & Moreno, 1999). IRT/CAT also features in many healthcare contexts because of its adaptation in Patient Reported Outcomes Measurement Information System (PROMIS) instruments (Cella et al., 2007; 2010; Segawa et al., 2020) that are widely used in FDA-regulated trials.

### 1.1 Item Response Theory (IRT)

IRT, a generative latent-variable modeling framework, models how a respondent of a given ability might respond to to each item in a testing bank. In IRT, an ability (canonically denoted $\theta$) is a theoretically continuous valued parameter (Bock et al., 1997; Immekus et al., 2019; Böckenholt & Meiser, 2017). The initial step for developing an IRT model involves creating a large pool of items that are topically grounded in a construct being measured. These items are then administered to a sizable and diverse sample of respondents, producing a dataset of item responses for model calibration. In the process of fitting an IRT model to the set of item responses, each item's specific parameters are determined (Kieftenbeld & Natesan, 2012; Luo & Jiao, 2018; Bürkner, 2021; Lord, 1983; Natesan, 2011; Natesan et al., 2016). Self-consistently, the ability for each of the respondents is also determined. Due to this coupling, the ability statistics for the calibration sample encode into the item-specific parameters, a fact made explicit by their relationship between IRT

and probabilistic autoencoders (Converse et al., 2019; Chang et al., 2019; 2023). Fundamentally, IRT maps each respondent's set of discrete item responses to a lower (usually single) dimensional latent space. In this manner, IRT models, like autoencoders, are nonlinear factorization models (Chang et al., 2021).

The goal of IRT is to apply such pre-trained models to new respondents, ranking them relative to the respondents used in model calibration. To do so, item parameters from calibration are held fixed and new responses for a given respondent are scored by solving an associated inverse problem for the ability parameter.

The possibly large item bank developed for the IRT model ideally has content coverage throughout the entire range of possible abilities. Administering a large item bank is burdensome for all parties involved. In the vicinity of any fixed ability parameter, however, the number of items is relatively small. CAT exploits this fact.

## 1.2   Computer Adaptive Testing (CAT)

The goal of computer adaptive testing (CAT) is to efficiently and accurately estimate a respondent's ability by using only the most relevant questions from a possibly large item battery. This selection is performed sequentially based on a running estimate of the respondent's ability. Selection methods mainly differ on the specific statistical objective being optimized. Generally, individual items are judged based on some measure of the degree to which they may improve the fidelity of the respondent's ability estimate. Most commonly, items are chosen greedily – while efficient, this type of selection procedure has the pitfall of poor item exposure.

Item exposure refers to the rate at which individual items in a testing bank are presented across multiple administrations. When exposure is poor, the effective instrument administered by the CAT is a limited subset of the items in the original bank. In unison with commonly-used improper scoring rules, this condition biases the resulting ability estimates. Having a small number of effective items also implies stereotypical item trajectories, making such instruments easier to game.

CAT methodologies select items based on a running estimate of a test-taker's ability. However, this estimate is unreliable at the beginning of the test, which in turn makes the statistical measures used to compare items noisy. For this reason, simply choosing the item that appears statistically best (a "greedy" approach) may not be ideal. A more effective strategy may be to hedge, selecting items that are useful across a wider range of potential ability levels.

In this manuscript we provide a methodology for hedging that is based on viewing item selection as a model selection problem. Each item implies a different model for the respondent's ability at the next step of the test. As a consequence of viewing the problem through these lens, we both motivate a new item selection criterion based on the information theoretic ability model discrepancy, and a stochastic selection procedure where the Frequentist statistics of item probabilities correspond to Bayesian model averaging statistics of the item-wise implied ability estimates.

## 1.3   Work Disability Functional Assessment Battery (WD-FAB)

As concrete tests of our methodology we used the eight independent IRT models, and their associated item banks, present in the WD-FAB (Meterko et al., 2015; Marfeo et al., 2016; 2019; Chang et al., 2022; Marfeo et al., 2018; Jette et al., 2019; Porcino et al., 2018). The WD-FAB characterizes whole body and mental function across four physical instruments and four mental instruments. The item banks consist of questions that ask about a range of everyday activities, such as emptying a dishwasher, walking a block, turning a door knob, speaking to someone on the phone, and managing under stress. Accepted responses were graded on either four or five option ordinal Likert scales.

The intended use of this instrument is to provide standardized and reliable information about an individual's functional abilities to help inform SSA's disability adjudication process. The WD-FAB provides eight scores across two domains of physical and mental function that are relevant to a person's ability to work.

As an application where item exposure is important, the eight independent models that comprise the WD-FAB are an ideal testing ground for our methodology.

## 2    Preliminaries

Suppose that one has developed a test bank consisting of $N_{\text{items}}$ items, of a given ordering, and used a set of responses to these items in order to calibrate an IRT model. The IRT model implies that a person of ability $\theta$ is expected to respond to the $i$-th item according to the probability mass function $p_i(k|\theta)$. For generality, we assume that the IRT model is polytomous so that there are $N_{\text{levels}}$ possible responses to each item. If a fully Bayesian approach was used in calibration, then $p_i$ can be the marginal probability mass found by integrating out the posterior item specific parameters. Otherwise, it is the probability mass implied by point estimates of the item specific parameters. For a given individual, knowing all of their responses $\boldsymbol{x} = (x_1, x_2, \ldots, x_{N_{\text{items}}})$, one may estimate the ability of the individual by computing the statistics of the posterior distribution

$$\pi(\theta|\boldsymbol{x}) \propto \pi(\theta) \prod_{i=1}^{N_{\text{items}}} p_i(x_i|\theta) \tag{1}$$

where the maximum likelihood estimate corresponds to using an uniform $\pi(\theta)$.

The objective of a testing session is to ascertain the ability of a new respondent, efficiently approximating the statistics of Eq. 1. In this sense, Eq. 1 is considered the *true estimate* of a person's ability. In computer adaptive testing (CAT), items are presented sequentially to a respondent. So, it is natural to base the choice of the next item on a combination of the current ability estimate and the properties of the ability estimate implied by the next item. In concrete terms, at step $t$ of a test, the items $\boldsymbol{\alpha}_t = (\alpha_1, \alpha_2, \ldots, \alpha_{t-1}, \alpha_t)$ have been answered by a respondent, from which a running ability estimate is obtained. Commonly, this estimate is based on statistics of the distribution

$$\tilde{\pi}(\theta|\mathbf{x}_t) \propto \pi(\theta) \prod_{s=1}^{t} p_{\alpha_s}(x_{\alpha_s}|\theta), \tag{2}$$

where $\mathbf{x}_t = (x_{\alpha_1}, x_{\alpha_2}, \ldots x_{\alpha_t})$ are the observed responses at step $t$. Then, the choice of item $\alpha_{t+1}$ is made conditional on this estimate. The item selector, conditional on the ability estimate, computes a given criterion for each of the remaining $N_{\text{items}} - t$ as a basis for making a decision.

### 2.1    Prior art

The oldest and perhaps most-popular CAT methodology is based on the principle of reducing the asymptotic variance of the ability estimate. This method chooses the item for step $t+1$, conditional on the point ability estimate at step $t$, $\hat{\theta}_t$ (commonly the expectation of Eq. 2), that has the maximum local item-wise *Fisher information*

$$I_i(\hat{\theta}_t) = - \frac{\partial^2}{\partial \theta^2} \sum_{k=1}^{N_{\text{levels}}} w_{ik} \log p_i(k|\theta) \bigg|_{\theta = \hat{\theta}_t}. \tag{3}$$

At step $t$ the CAT is not privy to the response to the next item so Eq. 3 requires a weighted sum over the potential responses. Typically one resolves the weights $w_{ik}$ self-consistently using the IRT model by setting them to $w_{ik} = p_i(k|\hat{\theta}_t)$, so that they compute an expectation (Magis, 2015), though sometimes uniform weights $w_{ik} = 1/N_{\text{levels}}$ are used. In this manuscript we will assume that the weights correspond to the former.

The Fisher information method while computationally expedient has several known limitations. First, the method adjudicates items conditional on the current running ability estimate. This quantity is not well-characterized early-on in an exam. A class of slight modifications to this criterion take ability uncertainty into account by computing an expectation of the Fisher information over a distribution of ability values (Owen, 1975; van der Linden, 1998; van der Linden & Ren, 2020; Ueno, 2013; Choi & Swartz, 2009):

$$\text{Bayesian Fisher information} = \int \tilde{\pi}(\theta|\mathbf{x}_t) I_i(\theta) \mathrm{d}\theta. \tag{4}$$

Information theoretic alternatives to the Fisher information are also motivated by resolving this issue, for example the *global information* method of Chang & Ying (1996),

$$\text{Global information} = \mathbb{E}_\theta \left[ \sum_k p_i(k|\theta) \log \frac{p_i(k|\theta)}{p_i(k|\hat{\theta}_t)} \right]$$

$$= \mathbb{E}_{x_i} \left[ \mathcal{D} \left( \tilde{\pi}(\theta|\mathbf{x}_t, x_i = k) \parallel \tilde{\pi}(\theta|\mathbf{x}_t) \right) \right] + \mathcal{D}_{x_i} [\tilde{p}_i^{(t)} \parallel p_i(k|\hat{\theta}_t)], \qquad (5)$$

where $\mathcal{D}(q(\theta) \parallel p(\theta)) = \mathbb{E}_{q(\theta)} \log[q(\theta)/p(\theta)]$, $\mathcal{D}_x(p(x) \parallel q(x)) = \sum_k p(k) \log(p(k)/q(k))$, and $x_i \sim \tilde{p}_i^{(t)}$ for

$$\tilde{p}_i^{(t)}(k) = \int p_i(k|\theta) \tilde{\pi}(\theta|\mathbf{x}_t) \mathrm{d}\theta. \qquad (6)$$

Other related information criteria that involve the KL divergence between the next ability estimate and the current ability estimate also exist (Sorrel et al., 2020; Wang & Chang, 2011; Weissman, 2007; Wang et al., 2020).

Second, the Fisher information provides an inaccurate approximation of the estimate precision when the number of observed items is small. Instead, one may directly compute the item-specific conditional variance (van der Linden, 1998)

$$\text{Bayesian variance} = \text{Var} \left[ \theta|\mathbf{x}_t, \alpha_{t+1} = i \right]. \qquad (7)$$

Third, greedy item selection methods have highly stereotypical item trajectories and poor item exposure. To address this issue, explicit and complex exposure controls exist (Georgiadou et al., 2007; Han, 2018), including by using randomness in the selection procedure Barrada et al. (2008), sampling items according to an ad-hoc function of the item-wise local Fisher information. Implementing these methods is challenging because they require tuning. The stochastic method for instance also relies on adaptive dampening of the sampling probabilities.

While various prior methods target reduction of the posterior estimate variance, they do not consider whether the posterior ability estimate is well-calibrated. Zhuang et al. (2023) introduced a gradient-based method where they select a subset of items that most closely matches the gradient of the likelihood function at an estimate of the true full-bank ability estimate.

## 2.2 Related methodologies outside of traditional CAT

CAT can be viewed as a particular application of Bayesian Optimal Experimental Design (BOED), which is a broad framework for choosing the next *experiment* or measurement for learning about a system based on maximizing a given utility (Rainforth et al., 2023). Unsurprisingly, many of the methods common to CAT have analogues in BOED, for instance in using Bayesian information theoretic criteria (Sebastiani & Wynn, 2000; Bernardo, 1979) or Frequentist experiment/itemwise Fisher information (Smith, 1918). The most common criterion in modern BOED is the expected information gain (EIG), and many stochastic methods for approximating this quantity exist (Laínez-Aguirre et al., 2015; Foster et al., 2020; Zaballa & Hui, 2023; Goda et al., 2022). However, unlike in CAT, there is not a strong motivation to use stochastic selection in order to improve exposure for BOED experiments.

## 3 Methods

Our main novel theoretical contribution is that we frame the CAT through the lens of model selection/model averaging, rather than directly as an optimization problem. At a given step $t$, the choice of the next item $\alpha_{t+1}$ is analogous to choosing among $N_{\text{items}} - t$ choices for the next ability estimate $\tilde{\pi}(\theta|\mathbf{x}_{t+1})$. If the full bank estimate were known then we could compute the item-specific ability model discrepancy measure

$$\mathcal{D} \left( \pi(\theta|\boldsymbol{x}) \parallel \tilde{\pi}(\theta|\mathbf{x}_{t+1}) \right) = \int \pi(\theta|\boldsymbol{x}) \log \frac{\pi(\theta|\boldsymbol{x})}{\tilde{\pi}(\theta|\mathbf{x}_{t+1})} \mathrm{d}\theta \qquad (8)$$

and use it as the basis for item selection. In particular, information theoretic model averaging techniques use the negative exponential of the model discrepancy to determine model weights (Akaike, 1978; Bozdogan, 1987; Dormann et al., 2018; Wagenmakers & Farrell, 2004; Yao et al., 2018). Our aim is to perform stochastic selection using these weights. However we first address the challenge of approximating Eq. 8, acknowledging that both distributions in the KL divergence are unknown at time $t$.

### 3.1 Plug-in estimation of the expectation of Eq. 8

Like in all CAT methods, we need to resolve our objective (Eq. 8) under incomplete observation. We pursue the usual strategy of computing an expectation. Computing the expectation of Eq. 8 exactly requires specifying $(N_{\text{items}} - t) \times N_{\text{levels}}$ different marginal posterior distributions, each of which is challenging to compute. In order to make the method tractable, we develop a mean field estimate of the expectation of Eq. 8. In this estimate, we ignore the coupling between $\pi(\theta|\boldsymbol{x})$ and the response to the next item, plugging in the expectation of $\pi(\theta|\boldsymbol{x})$, the marginal posterior,

$$\pi(\theta|\mathbf{x}_t) = \mathbb{E}_{\mathbf{z}_t} \pi(\theta, \mathbf{z}_t|\mathbf{x}_t), \tag{9}$$

into Eq. 8. In Eq. 9, $\mathbf{z}_t$ are the responses that have not yet been observed at step $t$. Still, the expectation in Eq. 9 is intractable. Fortunately, this expectation is amenable to Variational Bayesian Expectation Maximization (VBEM).

VBEM (Bernardo et al., 2003) allows us to iteratively approximate Eq. 9, producing a sequence of estimates $q_\theta^{(0)}, q_\theta^{(1)}, q_\theta^{(2)}, \dots$ that obey the descent property of the Majorization Minimization (MM) algorithm (Lange et al., 2000; de Leeuw & Lange, 2006; Lange et al., 2021; Wu & Lange, 2010) so that $\mathcal{D}(q_\theta^{(m+1)} \parallel \pi(\theta|\mathbf{x}_t)) \leq \mathcal{D}(q_\theta^{(m)} \parallel \pi(\theta|\mathbf{x}_t))$. Based on $q_\theta^{(m)}$, one can easily compute a corresponding set of response probabilities for all unobserved items. The VBEM update equations have the explicit form

$$\log q_{\mathbf{z}_t,j}^{(m+1)}(k) = \text{const}_j^{(m+1)} + \int \log p_j(k|\theta) q_\theta^{(m)}(\theta) \mathrm{d}\theta \tag{10}$$

$$\log q_\theta^{(m+1)}(\theta) = \text{const}^{(m+1)} + \log \pi(\theta) + \sum_{j \in \boldsymbol{\alpha}_t} \log p_j(x_j|\theta) + \sum_{j \notin \boldsymbol{\alpha}_t} \sum_k q_{\mathbf{z}_t,j}^{(m+1)}(k) \log p_j(k|\theta). \tag{11}$$

Then, after some number of EM iterations $M$, we can compute the plug-in criterion

$$\Delta_t^{(i)} = \sum_k q_{\mathbf{z}_t,i}^{(M)}(k) \mathcal{D}\left(q_\theta^{(M)}(\theta) \parallel \tilde{\pi}(\theta|\mathbf{x}_t, x_i = k)\right), \tag{12}$$

where $\mathcal{D}$ is the KL divergence. Technically, Eq. 9, rather than the commonly-used Eq. 2, is the best estimate of the ability at step $t$, an observation that we will save for the Discussion.

### 3.2 Stochastic item selector

It is our desire to hedge in the choice of the next item with frequency statistics that imply Bayesian model averaging (Hinne et al., 2020; Hoeting et al., 1999) of the corresponding per-item ability estimates. To do so, we draw the next item $i \notin \boldsymbol{\alpha}_t$, according to

$$\alpha_{t+1} \sim \text{Categorical}(\mathbf{w}_t) \qquad\qquad w_t^{(i)} = \frac{\exp\left(-\Delta_t^{(i)}\right)}{\displaystyle\sum_{j \notin \boldsymbol{\alpha}_t} \exp\left(-\Delta_t^{(j)}\right)}, \tag{13}$$

where the categorical distribution is defined over the $N_{\text{items}} - t$ items that have not yet been administered at time step $t$.

### 3.3 Relationship to cross validation

We can rewrite the discrepancy (Eq. 8) to remove the explicit dependence on $\tilde{\pi}$,

$$
\begin{aligned}
\mathcal{D}\left(\pi(\theta|\boldsymbol{x}) \parallel \tilde{\pi}(\theta|\mathbf{x}_{t+1})\right) &= \int \pi(\theta|\mathbf{x}) \log \frac{\tilde{p}_i^{(t)}(x_{t+1})\pi(\theta|\boldsymbol{x})}{p_i(x_{t+1}|\theta)\tilde{\pi}_t(\theta|\mathbf{x}_t)} \mathrm{d}\theta \\
&= \int \pi(\theta|\boldsymbol{x}) \log \frac{\tilde{p}_i^{(t)}(x_{t+1})}{p_i(x_{t+1}|\theta)} \mathrm{d}\theta + \mathcal{D}(\pi(\theta|\boldsymbol{x}) \parallel \tilde{\pi}(\theta|\mathbf{x}_t))
\end{aligned}
\tag{14}
$$

where

$$
\tilde{p}_i^{(t)}(k) = \int p_i(k|\theta)\tilde{\pi}(\theta|\mathbf{x}_t)\mathrm{d}\theta,
$$

and note that while the second term in the last line of Eq. 14 depends on the response for the next item, it does not depend on the choice of the next item. We can then relate the discrepancy to leave one out (LOO) cross validation, expanding the first term in Eq. 14

$$
\begin{aligned}
\mathcal{D}\left(\pi(\theta|\boldsymbol{x}) \parallel \tilde{\pi}(\theta|\mathbf{x}_{t+1})\right) &= \mathcal{D}(\pi(\theta|\boldsymbol{x}) \parallel \tilde{\pi}(\theta|\mathbf{x}_t)) + \int \pi(\theta|\boldsymbol{x}) \log \frac{\tilde{p}_i^{(t)}(x_i)\pi(\theta|\boldsymbol{x})}{\pi(\theta|\boldsymbol{x})p_i(x_i|\theta)} \mathrm{d}\theta \\
&= \mathcal{D}(\pi(\theta|\boldsymbol{x}) \parallel \tilde{\pi}(\theta|\mathbf{x}_t)) + S[\pi(\theta|\boldsymbol{x})] - \mathcal{D}(\pi(\theta|\boldsymbol{x}) \parallel \tilde{\pi}(\theta|\boldsymbol{x} \setminus \{x_i\})) + \log \frac{\tilde{p}_i^{(t)}(x_i)}{\tilde{p}_i^{\mathrm{LOO}}(x_i)}
\end{aligned}
\tag{15}
$$

where, $\tilde{\pi}(\theta|\boldsymbol{x} \setminus \{x_i\})$, the ability estimate when leaving out $x_i$ follows Bayes rule, $p_i(x_i|\theta)\tilde{\pi}(\theta|\boldsymbol{x} \setminus \{x_i\}) = \pi(\theta|\boldsymbol{x})\tilde{p}_i^{\mathrm{LOO}}(x_i)$ and $\tilde{p}_i^{\mathrm{LOO}}(x_i) = \int p(x_i|\theta)\tilde{\pi}(\theta|\boldsymbol{x} \setminus \{x_i\})\mathrm{d}\theta$ is the corresponding LOO mass function for item $i$. In this representation, only the last two terms in Eq. 15 depend on the item choice. So in minimizing the discrepancy, one is also selecting the item that if left out would yield the biggest discrepancy.

### 3.4 Numerical implementation

We coded two independent implementations of our methodology as applied to the Graded Response Model: one in Python (redacted) and one in Golang (redacted). Within our implementation we approximated all integrals using trapezoid approximations with 200 equally spaced grid points. We used $M = 5$ iterations to approximate the marginal posterior distributions (Eq. 11).

## 4 Results

In producing the following results, for each scale, we simulated item responses for 500 respondents for each true underlying ability of $\theta \in \{-3, -2.5, -2, \ldots, 2.5, 3\}$. Then we put each respondent's item responses through each CAT item selection method, obtaining ability estimates at given test lengths. The methods evaluated are greedy selection via the Fisher Information (Eq. 3), Bayesian Fisher information (Eq. 4), Global information (Eq. 5), Bayesian variance (Eq. 7), ability estimate discrepancy (Eq. 12), and our stochastic selection method (Eq. 13). Finally, we also computed ability estimates for each simulated respondent based on all of their item responses. In the main text we report on only the four mental scales of the WD-FAB. Please see the Supplement Results for the corresponding physical scale results.

### 4.1 Testing error

Figures 1, 2 and 3 provide different measures of ability estimation error in the context of computer adaptive testing. Fig. 1 displays values of the discrepancy (Eq. 8) conditional on the scale, item selection method, test length at stopping (5, 10, 20, 30, 40 items), and true fixed ability used in simulating CAT responses. Using the Fisher information and global information selectors, there are some situations in which the discrepancy increases as the test length increases for an intermediate range of test lengths before dropping. On the other hand, the Bayesian variance and the methods based on our criterion (Eq. 12) reliably decrease the discrepancy as the test length increases. Failure to decrease this discrepancy suggests that a selection

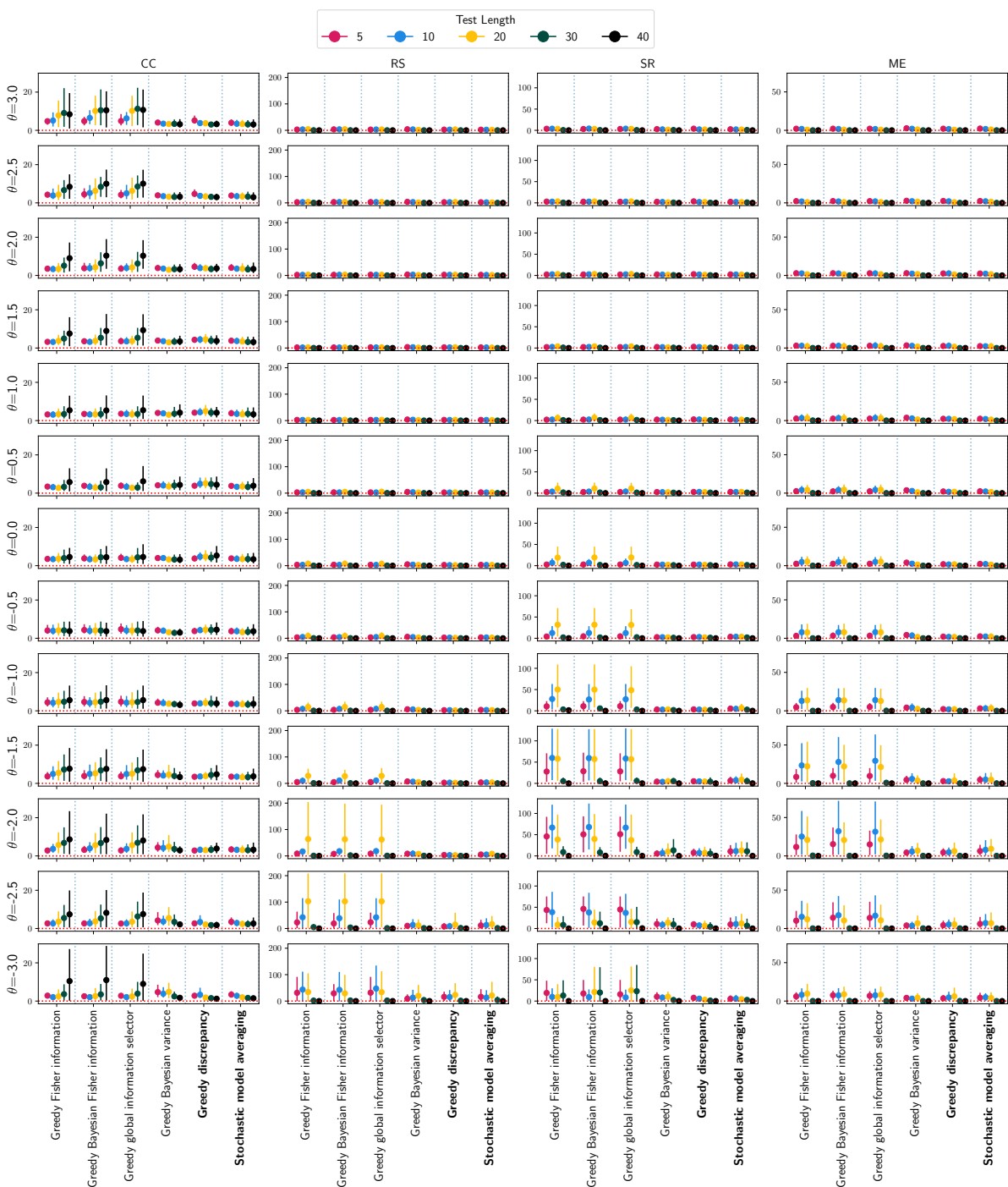

Figure 1: **Ability estimate discrepancy** $\mathcal{D}(\pi(\theta|\mathbf{x}) \parallel \tilde{\pi}(\theta|\mathbf{x}_t))$ (mean and middle 80% interval) conditional on score $\theta$ used to generate response sets, by scale, item selection method, and test length $t$, for mental function scales of the WD-FAB. Lower is better.

procedure generates item subsets that provide inaccurate ability estimates when used as whole-distribution A/B comparisons between individuals because those estimates are ill-calibrated.

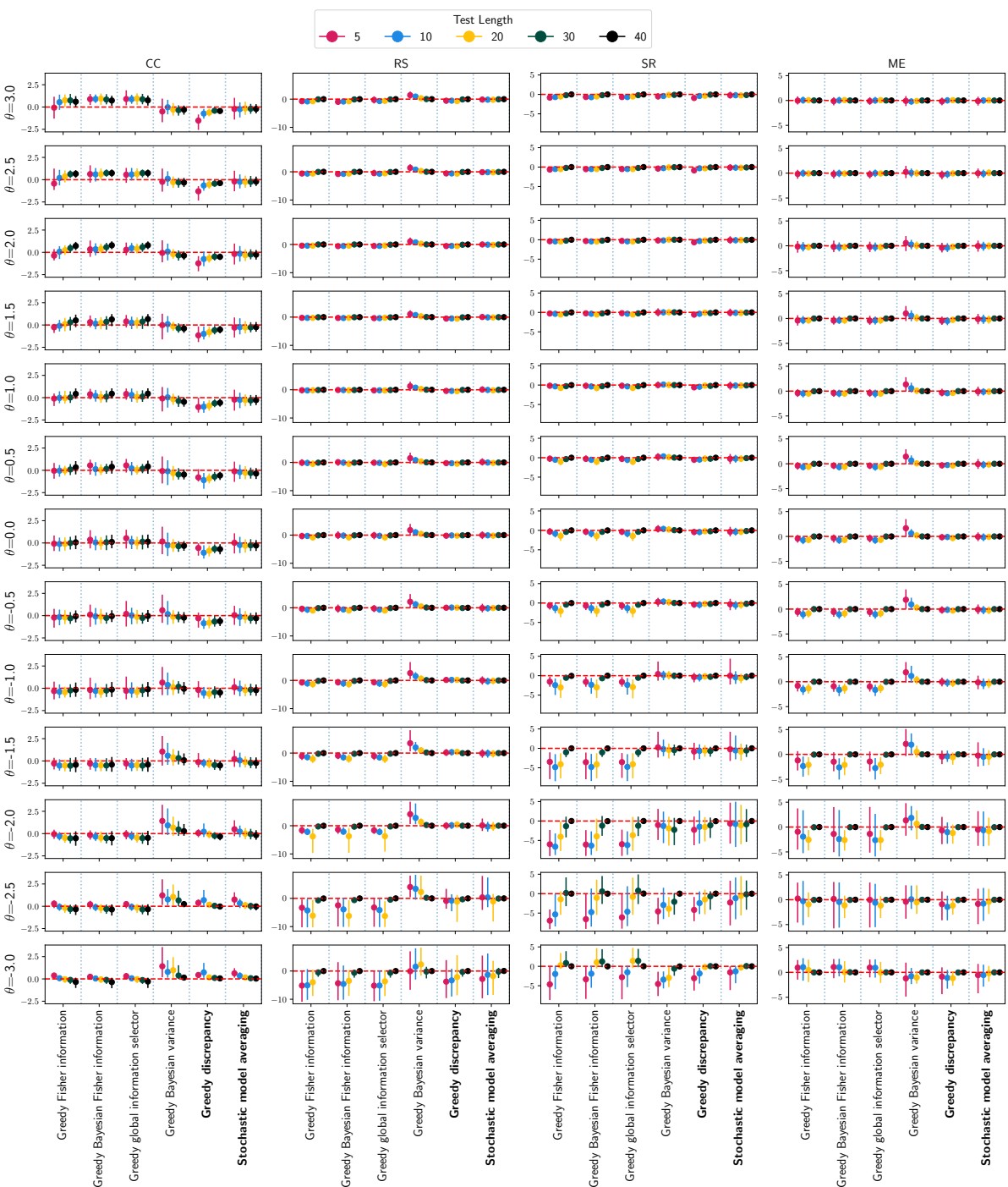

Figure 2: **Error in means** $\left(\int \theta \tilde{\pi}(\theta|\mathbf{x}_t)\mathrm{d}\theta - \int \theta \pi(\theta|\mathbf{x})\mathrm{d}\theta\right)$ (mean and middle 80% interval) conditional on true score $\theta$ by scale, item selection method, and test length $t$, for mental function scales of the WD-FAB.

In many CAT/IRT based instruments, the mean ability is used in order to characterize a respondent. Fig. 2 presents statistics of the mean ability error (mean and middle 80% coverage) across the different simulation configurations. In Fig. 3, we provide statistics of the absolute value of this error across simulations.

The error distributions are highly variable across these attributes. Generally, the magnitude of the error decreased as the test length increased. For most scales, there is a region of abilities for which all item

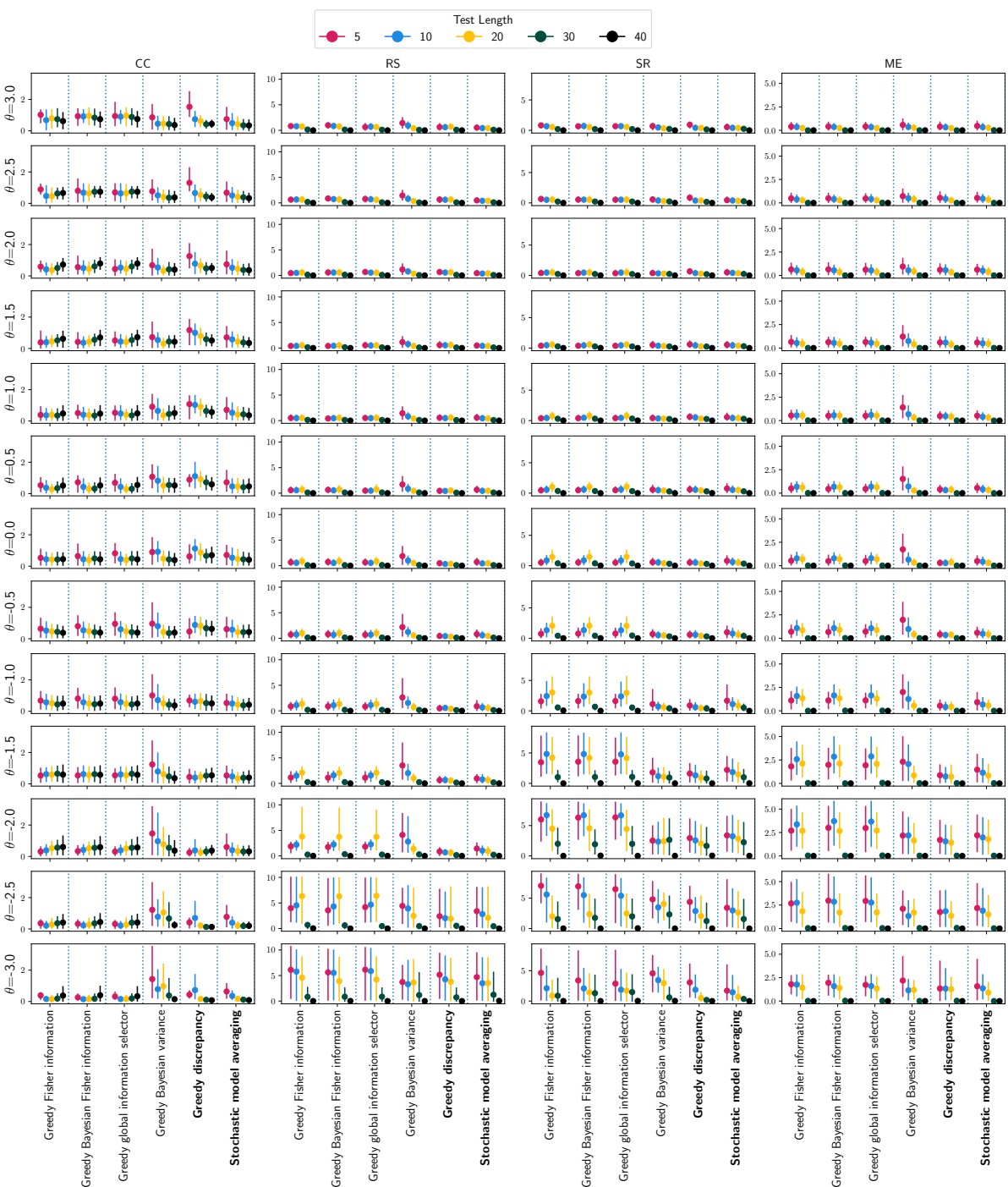

Figure 3: **Absolute error in means** ($|\int \theta \tilde{\pi}(\theta|\mathbf{x}_t)\mathrm{d}\theta - \int \theta \pi(\theta|\mathbf{x})\mathrm{d}\theta|$) (mean and middle 80% interval) conditional on true score $\theta$ by scale, item selection method, and test length $t$, for mental function scales of the WD-FAB. Lower is better.

selectors produced small errors. No single selection method had the lowest errors in all situations, though generally the stochastic selector performed most-consistently well.

Often, the posterior variance is used to define a cutoff for a CAT stopping rule. The standard deviation of the posterior ability estimates is presented in Fig. 4 for the different simulation configurations. In these

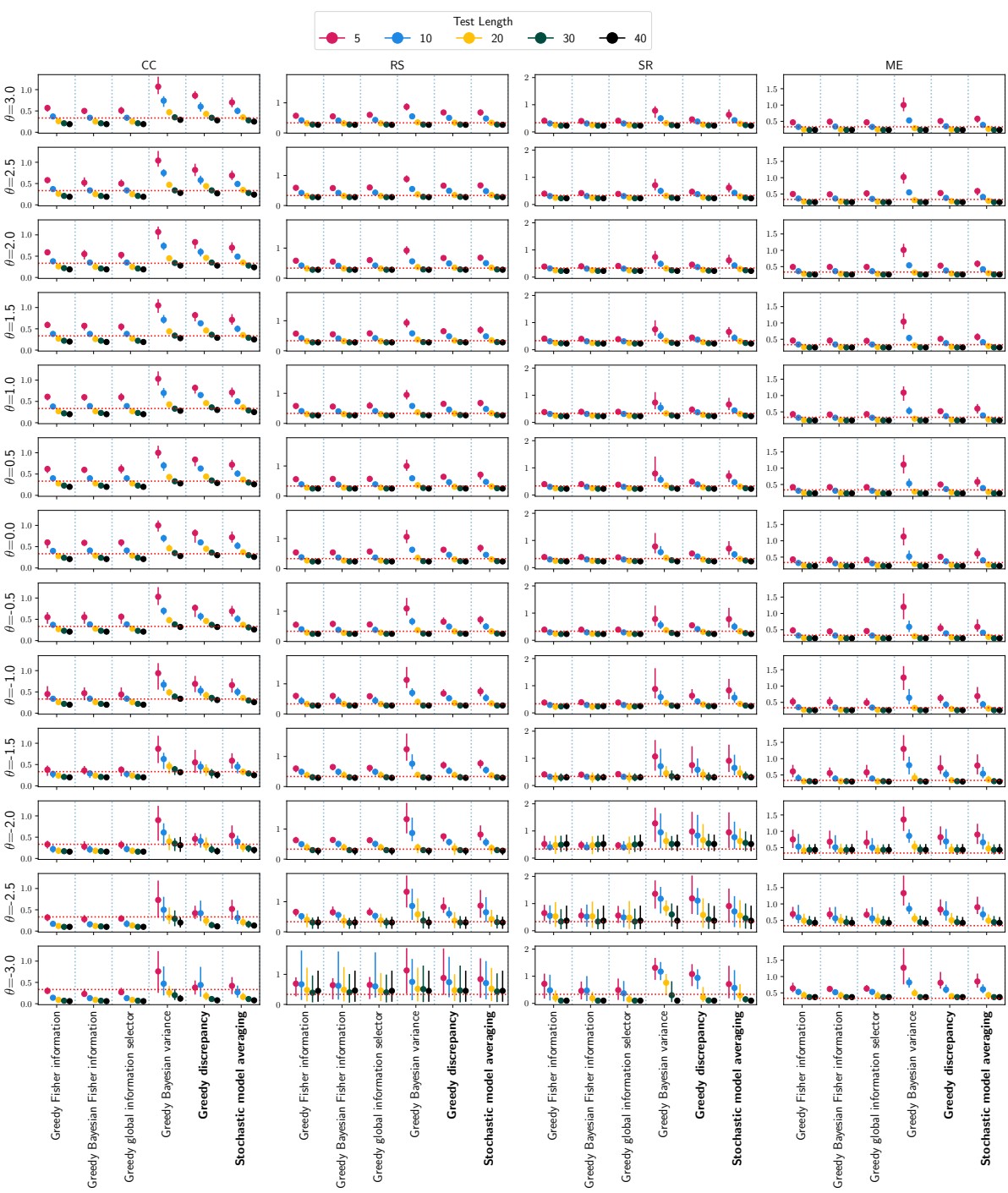

Figure 4: **Standard deviation of ability estimates ($\sqrt{\mathbf{Var}_t(\theta)}$** (mean and middle 80% percentile) conditional on true score $\theta$ by scale and item selection method, for mental function scales of the WD-FAB. Used as stopping criteria for CAT. Lower is better.

simulations, it is clear that the two Fisher methods and the global information method provide the lowest posterior ability standard deviations. However, in light of Figures 1, 2, 3, it is clear that these ability estimates are ill-calibrated. They are terminating quicker than they should and settling on sub-optimal ability estimates.

## 4.2 Item exposure

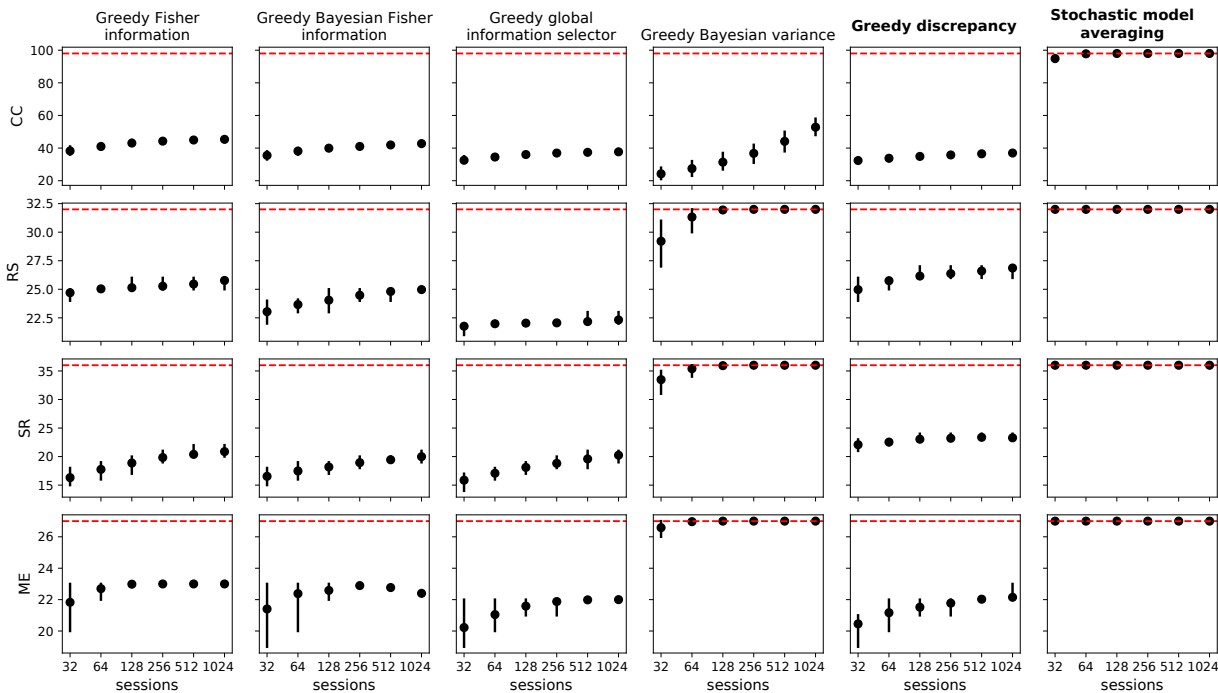

Figure 5: **Item exposure statistics** (mean and middle 80% interval), for each of the given item selection methods across a given number of CAT sessions, for mental function scales of the WD-FAB. The dashed line represents the maximum possible exposure per scale. Higher is better.

Fig. 5 compares the different item selection methods on the basis of item exposure across sessions (with 12 items presented per scale) with randomly distributed abilities. In this figure, for each simulation configuration, we counted the number of unique items seen for each scale across replications of the given number of CAT sessions. For example, for the scale "ME," we estimate that in each set of 32 sessions approximately 22 items are exposed on average, though with wide variance. As the number of sessions increases, the number of exposed items increases. Of the greedy methods, the Bayesian variance method has the best item exposure. For some scales, the Bayesian variance method performed almost as well as the best selection method, the stochastic selector based on Eq. 13. The stochastic selector successfully exposed all items for all scales in all the scenarios tested.

## 5 Discussion

In this manuscript we have introduced stochastic selection for CAT where the frequency statistics of the next item correspond to Bayesian model averaging of corresponding discrepancy weighted next item-wise ability estimates. In formulating our method we identified the ability discrepancy (the KL divergence between the next item ability estimate and the full bank true ability estimate) as a selection criterion. We provided a computationally expedient plugin version of our criterion based on variational Bayesian expectation maximization. Using simulations of the new selector (and other selectors for comparison), on the WD-FAB, we found our new stochastic selector to have both superior item exposure properties while not compromising in

terms of accuracy. Additionally, the simulations showed that unlike the Fisher information methods, the new selection methods (whether greedy or stochastic) are not over-confident in estimating scoring error. This fact implies that the new methods are less likely to settle on a poor ability estimate. Beyond characterizing a point estimate for ability, using the discrepancy as a criterion optimizes the whole-distribution ability estimate, which implies more-accurate A/B tests when comparing scores between different respondents. Finally, the computationally expensive portion of our overall approach is in computing the marginal posterior ability estimate. As we will discuss, this quantity is the true ability estimate at step $t$ and should be computed and used in all other selection methods. For this reason, our criterion is of similar computational complexity to the other Bayesian criterion mentioned in this manuscript.

## 5.1 What should the ability estimate be at step $t$?

In formulating our method, we assume that one is using a scoring methodology similar to what is commonly used in the literature – using the likelihood of the items observed up to step $t$. Recall that we call the posterior ability estimate obtained by this method $\tilde{\pi}(\theta|\mathbf{x}_t)$, making a distinction between this quantity and $\pi(\theta|\mathbf{x}_t)$, the marginal posterior ability at step $t$. The latter estimate differs from the former in that it also accounts for the fact that the $N_{\text{items}} - t$ unobserved items at time $t$ will also impact the final ability estimate. The latter is a better estimate of the ability because it is consistent with both the observed and unobserved items being drawn from the same underlying conditional distribution. For this reason, it should also be used in all selection methods in place of $\tilde{\pi}$ when taking expectations over unknown responses and in both the running and final score estimates. In a follow-up to this manuscript, we will elaborate on this point.

## 5.2 Why ensembling?

Focusing on efficiency, there are reasons to think why randomization in CAT would be sub-optimal. If the objective is to optimize a given criterion, then not always choosing the exact optimal item would seem to result in a less efficient CAT. As we have shown for the WD-FAB, this assumption did not hold. On the other hand, there are at least a couple a-priori explanations in support of our findings. First, in the context of prediction, Le & Clarke (2022) has shown that model averaging is asymptotically better than model selection. Second, each criterion requires resolving unknown future responses. Since the true ability of the respondent is unknown, the statistics of these responses is unknown. However, our method uses the *correct* item response probabilities in computing the expectation in Eq. 12.

## 5.3 Limitations and extensions

In using the variational Bayesian EM estimates for the marginal item probability mass functions in order to compute the item-wise expectations of Eq. 12, we are using the optimal item probabilities provided by the given IRT model. However, one may also be able to improve the accuracy of this expectation by using different IRT models that are more-tuned to accuracy than interpretability (Chang et al., 2019; 2023), so long as one accounts for unobserved items.

The estimate of the criterion of Eq. 8 in the form of the the mean field plugin estimator in Eq. 12 trades accuracy for computational efficiency. One could more-accurately compute this expectation by developing a version of Eq. 12 that preserves the coupling between $\pi(\theta|\mathbf{x})$ and the response to the next item.

This work was focused on improving the assessment of the WD-FAB, a factorized multidimensional IRT model. We found generally, across all scales (dimensions) that our model ensembling stochastic selector outperformed the other commonly used selection methods that we tested. Your mileage my vary when trying these methods with other instruments.

While we formulate our methodology assuming a multidimensional ability parameter $\theta$, it would likely take additional work in order to adapt this method to non-factorized multidimensional instruments. Additional controls might be needed in order to balance out the administration of the different scales for instance.

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
