# OpenReview forum: "Bayesian information theoretic model-averaging stochastic item selection for computer adaptive testing"
_TMLR — Rejected by TMLR_

### Review · Reviewer_xBpm · 2025-06-06

**Summary Of Contributions:**

This paper focuses on sequentially recommending items (or questions) in the context of computer adaptive testing. The authors propose using the gap between two Bayesian information measures: one estimated assuming all the items are presented, and the other estimated using the past responses along with one item from the unselected pool. An informative item is defined as the one that minimizes the Bayesian information. In addition, the authors propose a stochastic item selection strategy to encourage exploration of unselected items. They present simulations comparing the proposed method with multiple baselines.

**Audience:**

Yes

**Broader Impact Concerns:**

I did not find any.

**Claims And Evidence:**

No

**Requested Changes:**

**Abstract and Introduction**\
I recommend that the authors revise both the abstract and the introduction to enhance clarity. For the abstract, consider using layman-friendly language to make the core ideas more accessible to a broader audience. For the introduction, I suggest clearly explaining what computer adaptive testing (CAT) is, outlining the key challenges associated with it, and describing how this paper addresses those challenges. Additionally, the use of subsections within the introduction is unnecessarily detailed and makes the reading experience less fluid; a more concise and focused narrative would be more effective.

**Preliminaries Section**\
Please consider adding a dedicated preliminaries section that introduces the mathematical foundations of CAT.

**Section 2 (Methodology)**\
I suggest the authors begin Section 2 with a high-level description of the Bayesian framework for item selection. Furthermore, each subsection in Section 2 should clarify their motivations rather than diving directly into the proposed solutions.

**Section 2.4 – Innovation and Justification**\
The innovation presented in Section 2.4 is currently not well elaborated and appears somewhat ad hoc. If this work is meant to be primarily an application of existing techniques, the authors should provide a clear justification for using Bayesian information in this specific context. This includes a comparison with other acquisition functions (e.g., Fisher information) and an explanation of why the proposed stochastic selection strategy adds value.

**Strengths And Weaknesses:**

## Strength ##
1. Selecting informative items to improve the efficiency of computer adaptive testing (CAT) is practically valuable.
2. The method used to identify informative items is reasonable.

## Weaknesses ##

**1. Methodological Contribution**\
The innovative aspects of this work are not clearly presented. Using Bayesian information as a criterion for selecting informative items is a standard approach in active learning. Therefore, the novelty appears to reside primarily in the stochastic item selection strategy proposed in Section 2.4. However, the authors do not provide a clear justification for why this strategy is necessary or beneficial. Overall, the proposed methodology lacks originality and is similar to techniques already applied in various label-efficient settings. If this work is intended as an application paper rather than a methodological contribution, the authors should justify why Bayesian information is an appropriate acquisition function, especially when compared to alternatives such as Fisher information.

**2. Presentation**\
The introduction feels redundant, particularly due to the inclusion of subsections. The authors should consider focusing the introduction on clearly explaining, in plain language, what CAT is, what challenges it presents, and how their methodology addresses those challenges. Several sentences are difficult to read. Examples include:\
*“The objective of computer adaptive testing (CAT) is to tailor the administration of an item battery to the ability on the respondent so that one can obtain a precise estimate for the respondent’s ability in a shorter amount of time than that required to administer the entire battery.”*

*“We formulate the optimization problem for CAT in terms of Bayesian information theory, where one chooses the item at each step based on the criterion of the ability model discrepancy – the statistical distance between the ability estimate at the next step and the full-test ability estimate.”*

*“The objective of a testing session is to use this model to ascertain the ability of a given individual relative to that of the calibration sample, approximating the statistics of Eq. 3 in as efficient a manner as possible.”*

**3. Lack of Clarity in Methodology**\
Section 2 presents the methodology in a way that instructs readers on what is being done without offering sufficient clarity. For example, in Section 2.2, the authors state:*“This issue is not unique to our methodology, and is usually resolved by taking an expectation using a given mass function (typically using the current ability estimate).”* It seems that this refers to taking an average over results of possible responses to the remaining items—this can be stated explicitly  formulated in an equation. Furthermore, Section 2.3 introduces an EM algorithm to estimate $\theta$, but the authors do not clarify its role or purpose. Lastly, the benefits and motivation for the stochastic item selection described in Section 2.4 are not well articulated.

---

> ### Author Response · Authors · 2025-06-09
> **Placeholder of a reply**
>
> Thank you so much for your prompt review and helpful suggestions. I'm going to hold off on detailed edits to the manuscript and a more-specific reply until we get a couple more reviews in. I'll edit this comment in the next week or two. In the meantime I will look through work in the active learning field that potentially relates to this manuscript. I'm also slowly going over the manuscript and considering ways to simplify the writing. Stay tuned and thanks again.

---

> > ### Author Response · Authors · 2025-06-25
> > **Response to xBpm**
> >
> > Openreview won't let me edit my previous placeholder so I'm posting a new reply.
> >
> > IRT + CAT (or variants/extensions of CAT) are dominant methods behind assessments for using a quiz in order to obtain an ability (also called trait) estimate $\theta$. For instance, the FDA/NIH has published a set of guidelines (called PROMIS) for both IRT and CAT. One of our objectives is to update these guidelines to remove a particular blind spot:
> >
> > When using a deterministic greedy selection procedure for CAT, one gets very stereotypical item trajectories and effectively only a small fraction of an item bank is every served. This fact, coupled with using the commonly used (bad) convergence criterion of the posterior variance (like in Fig 4) means that one gets biased ability estimates that are inconsistent with what the full item bank would give for a respondent. This is made much worse by the fact that it is common to use the incorrect Eq (4) rather than the correct Eq (7) in scoring a respondent without knowing all of their responses.
> >
> >  In this manuscript, we are providing a new methodology for performing the stepwise item selection that is based from the ground up on the principle of model averaging across the item-wise posterior ability posterior candidates based on their deviation from the unknown true ability posterior. This principle leads to stochastic selection: the item sampling stats correspond in the frequentist-sense to model averaging in the Bayesian sense of the possible next ability estimates. Collectively these methodologies are novel. This methodology is also principled; I do not believe that it is ad-hoc.
> >
> >  It is coincidental (and mathematically interesting in our view) that the natural criterion has similarities to certain more-modern information criteria used in the CAT literature. However, previous literature has not explored the mathematical similarities and differences, which we did in Section 2.5.
> >
> >
> >
> > I'm still working on paper edits in response to your and the others' reviews - I will post a new revision over the weekend. In the meantime here are the changes that I am making:
> >
> > * I am making an effort to simplify the writing a la Hemingway.
> > * I am reorganizing the manuscript, putting it into a more-traditional format common for ML papers, making sure also that each section starts with a short blurb on motivations.
> > * The EM algorithm is used for computing the correct posterior ability estimate under incomplete observations. It is not the only way to do so but it reliably converges and has simple closed form updates. We'll better motivate it in the revision.

---

> > > ### Author Response · Authors · 2025-06-25
> > > **Response to xBpm re: active learning**
> > >
> > > I forgot to mention that we'll also comment on active learning/bayesian optimal experiemental design methodologies and their relationship to CAT in the updated manuscript.

---

### Review · Reviewer_ifxV · 2025-06-17

**Summary Of Contributions:**

The authors propose a variational Bayes criterion for selecting test items that maximize the information about latent person ability parameters as conceptualized within item response theory (IRT). The additional stochastic selection procedure casts item sampling as Bayesian model averaging over ability estimates using adaptive weights. While the approach appears to offer an interesting contribution, the paper’s poor structure and readability make it very difficult to fully assess its significance and the extent of the empirical evaluation. I am left wondering if the paper is not more suitable for a specialist journal on educational measurement.

**Audience:**

Yes

**Broader Impact Concerns:**

No concerns.

**Claims And Evidence:**

No

**Requested Changes:**

- There is ambiguity regarding local Fisher information (Eq. 1) and Fisher information generally (defined as the variance of the score). The authors sometimes refer to the latter when having the former in mind. In this regard, I would advocate for a less notation-heavy Introduction that clearly contextualizes the work and states the contributions, and a much-improved Notation / Background section. It is very strange that lots of (somewhat irrelevant) notation currently comes prior to the actual **Notation** section.
- Why is there a complete disconnect between CAT and (adaptive) Bayesian optimal experimental design (BOED)? The literature on BOED has exploded in recent years (see, e.g., the excellent review by [1]) and there are many methods going beyond mean-field VI. I may be wrong, but Bayesian CAT seems to be a special case of adaptive BOED. Please, consider including a Related Work section that clearly puts the current paper in the context of Bayesian IAT models, CAT, and adaptive optimal design.
- The **Notation** section itself needs lots of improvement, as it currently leaves a lot to the reader's imagination and leverages unintuitive abuse of notation. Since the IAT likelihood is assumed to be exchangeable, there is no difference between Eqs. 3 and 4. except that Eq. 4 considers partial (exchangeable) history. Why is the nested subscript $\alpha_s$  needed? Why do sections **2.1** and **2.2** take up so much space to explain simple Bayesian updating?
- Most terms in Eqs. 8 -10 are undefined. Eq. 12 has a problem, as $\tilde{p}^{(t)}_i(k)$ does not appear in Eq.12. Again, there are undefined terms.
- The role of **Section 2.6** is unclear, as it is completely disconnected from the flow of the Methods section. Please, motivate.
- Why is marginal posterior standard deviation a good metric (Figure 4)? As a Bayesian, one should be much more concerned about calibration (especially when using cheap posterior approximations like mean-field), since contraction alone can easily be gamed. Please, streamline the entire Methods section and make sure all terms and distributions are properly defined, as well as all approximation assumptions clearly stated (e.g., mean-field is all about factorial Gaussian approximation).
- Figure 5 and the related conclusion needs an explanation.
- IAT models are typically estimated in a hierarchical Bayesian modeling framework. Is the method applicable to this most common setting?

[1] Rainforth, T., Foster, A., Ivanova, D. R., & Bickford Smith, F. (2024). Modern Bayesian experimental design. Statistical Science, 39(1), 100-114.

**Strengths And Weaknesses:**

**Strengths**
- The paper addresses an interesting question and seems relevant for model-based CAT.
- The empirical evaluation is comprehensive and looks at different metrics and methods.
- The authors openly acknowledge limitations of the proposed mean-field method.

**Weaknesses**

- The paper is poorly written and suboptimally organized, following a highly atypical outline for an ML paper. At the start of **Section 2**, it is still unclear what the contribution of the paper is and how it should be contextualized within the related work scattered across the intro subsections full of implicit notation. In fact, I even had a hard time connecting the title to the paper.
- The background on IRT is very limited and does not clearly articulate the various IRT model formulations in the Bayesian literature (see, e.g., [1, 2, 3]. Thus, throughout the entire paper, the reader is left wondering about which model parameterization was used, how were typical problems of IAT models (e.g., identifiability) resolved, and what happened to item difficulty parameters that are typically jointly estimated with ability parameters.
- The results section is nearly impossible to follow or connect to the Methods. Are the differences between the greedy methods and stochastic MA noteworthy or practically relevant? How does one make sense of the differences between data sets. Why are mean differences and marginal variance considered good metrics in a Bayesian context (see also points below)?
- Only one IAT model parametrization is used throughout the evaluation (if I understood the results section correctly), which raises questions about the generalizability of the results, especially to non-factorial or hierarchical models.

[1] Bürkner, P. C. (2021). Bayesian item response modeling in R with brms and Stan. Journal of statistical software, 100, 1-54.
[2] Bürkner, P. C., Schulte, N., & Holling, H. (2019). On the statistical and practical limitations of Thurstonian IRT models. Educational and Psychological Measurement, 79(5), 827-854.
[3] Böckenholt, U., & Meiser, T. (2017). Response style analysis with threshold and multi‐process IRT models: A review and tutorial. British journal of mathematical and statistical psychology, 70(1), 159-181.

---

> ### Author Response · Authors · 2025-06-25
> **Response to ifxV**
>
> Thanks for your detailed review. I appreciate that you caught several typos. In response to your review and that of xBpm I will reorganize the manuscript to put it into an ordering that is more-traditional of ML manuscripts. That will address some of your issues wrt out of order notation usage etc. I will post an updated manuscript over the weekend. In the meantime, addressing your comments:
>
> - BOED: You are correct that CAT is linked to BOED. Our methodology also extends to BOED but that would widen the work beyond the intended already-large scope of the paper. IRT/CAT as we have mentioned is the *dominant* modeling paradigm for determining an ability estimate from a set of items. We'll link CAT to BOED (and active learning in response to the other reviewer) in the new related works section. As a side note, I did a literature search for stochastic BOED methods, finding recent work on stochastic gradient based BOED. Interestingly, without knowing about this work I initially had two separate paths for viewing CAT/stochastic selection: 1) model selection/averaging which I settled on 2) stochastic optimization. I haven't found BOED methodology that makes the model selection/averaging -> stochastic selection analogy. I also am not aware of work in CAT that takes advantage of this newer methodology in BOED.
> - Notation: In reordering the manuscript I'll put the equations *after* a notation section. Some of the notational stuff that was confusing to you was in the use of superscripts which I'll make sure I explain in the updated manuscript
> - Exchangeability: yes exchangeability holds but that's not relevant to Eq (4), because crucially t<I. Under partial observation, Eq (4) does not provide the correct likelihood function for the data. The correct likelihood is Eq (7) which takes into account the fact that I-t responses are missing after receiving the t-th response.
> - Nested subscript: The items already have a given ordering. The second subscript indexes against the original item ordering because we are giving the items out of order.
> - Bayesian updating: I am debating cutting Eq (6) or moving it to the Supplement. I like how the manuscript is currently didactic and doesn't skip any crucial steps in deriving the criterion.
> - Section 2.6: We are attempting to 1) introduce the other commonly used criteria 2) show how the new criterion is both related an unrelated. That said, we will turn this section into a related prior works section where we also will note some BOED/AL methods.
> - Marginal posterior standard deviation (Figure 4): It's not a good metric! We need to emphasize that more. The problem is that most of the prior literature is using that as a metric, and in-the-wild, using that in order to determine convergence of the CAT.
> - Mean field: We will make this more clear in the updated manuscript. The VBEM ability estimate is **not** mean-field. It is what you get by the standard EM-MM updating when you optimize KL(q_theta || E _z pi(theta, z| x)). There is no mean field in the ability estimate.
>
> Where we use mean field is in Eq (10) which is is a mean-field approximation of Eq (5). We note that we do this for practical reasons because the computational difference is large. If not using mean field then you need to run the EM procedure (I-t)xK times for each missing item and its possible responses. We'll note that.
> - IRT/hierarchical modeling: Yes, we originally had a more general formulation of IRT in the text but removed it in order to  simplify the exposition. In the general Bayesian sense you would marginalize over all hyperparameters to get an effective (marginal) likelihood function. Note that while our experiments are on the graded response model, the actual form of the IRT model is not important, all the equations in our text still apply. Actually the ability model doesn't even need to be IRT. It can be any model that has a latent ability parameter that you can apply to new responses in order to infer an ability estimate for the new responses.

---

> > ### Author Response · Authors · 2025-06-25
> > **Response to ifxV part 2)**
> >
> > To address the previous noted weakness on not having enough background on IRT, we will expand on section 1.2 and cut some text out of section 1.3. Crucially, this is how IRT models are created and then used:
> >
> > Calibration: A set of responses from many people are fitted to the IRT model. As a result of this procedure you get the item specific parameters AND the ability estimates for the set of people used for calibration. The IRT model latent ability parameter is an effective rank transformation on the abilities of the people at calibration. Since one is using real data, missingness can occur.  In the presence of missingness, it is common still for people to use the incorrect Eq (4) *except* I recently found a manuscript that I will cite that uses the correct Eq (7) + variational stochastic gradient descent. I am also working on a separate paper where I provide an MM algorithm.
> >
> > Scoring (as in CAT): The previously trained model is applied to new people by freezing the item parameters and inferring the new person-wise statistics. This is usually done using the *incorrect* Eq (4) rather than the correct Eq (7).

---

> > > ### Comment · Reviewer_ifxV · 2025-07-30
> > >
> > > First off, I would like to apologize for my delayed response. I thank the authors for the comprehensive revision, which addressed many of my points. I would still ask for a thorough sanity check of the paper, as some typos and notational inconsistencies remain:
> > >
> > > - Both $\boldsymbol{x}$ and $\mathbf{x}$ are used to denote the vector of observed responses.
> > > - I really don't see the need for the nested index $\alpha_s$ denoting the items. The items are after all, just a set of integer indices and it may be much simpler to write eq.2 in terms of the respective factorization rather than indexing a distribution or a set of RVs using a set of indices. Also, the latter violates the practice in the paper of using bold font for vectors (i.e., $x_{\alpha_{s}}$ is already a vector at $s =2$).
> > > - The unpacking of the definition of global information (Eq. 5) seems unnecessarily verbose. Eq. 6 is not used in any downstream derivation.
> > > - The " item-specific ability model discrepancy measure" is simply the Bayesian surprise or information gain.
> > > - I still think the paper will profit from a short list of the main contributions the authors make before diving into the **Methods** section.
> > > - Eq. 15, integral on the RHS -> are all terms correct?
> > > - There are no references to the appendix in the main text.
> > >
> > > Additionally, the enormous results figures remain difficult to read, as they have lots of redundant information. What does the bold font stand for? Best performance across rows? Usually, this would be a table and you would highlight the best performing method according to a given metric in bold. Showing both mean differences and absolute mean differences seems redundant. You could just report RMSE. In a Bayesian framework, you should be concerned with calibration of the credible intervals (which can be tested via simulation-based calibration and quantified via ECE) in addition to accuracy and contraction.

---

> > > > ### Author Response · Authors · 2025-07-30
> > > > **Re: ifxV**
> > > >
> > > > Thanks for your diligence in checking over the manuscript. I'll think your recommendations are good but some of them will take some time in order to implement...
> > > >
> > > > This question is more for the editor but can we have a couple weeks to post another revision? The original timeline for review/decision seems to be off anyways.
> > > >
> > > > Some quick specific responses:
> > > > - $\boldsymbol{x}$ is used to denote the vector of responses if all items in the bank are observed. $\mathbf{x}_t$ is used to denote the vector of the limited number of observed items at step $t$
> > > > - I'll take the index comment into consideration
> > > > - Eq (5) was mainly intended to point out how the two methods (global information) and ours might be similar or different. We included this mainly because the global information is presented as it is in the first line of Eq (5), as a KL divergence over the discrete space of responses. We wanted to convert that to KL divergence in ability space. I'll think about a way of better motivating Eq (5)'s inclusion
> > > > - I've checked Eq 15 over before, I can add one or two more lines of the proof into the appendix (and simplify what is in the main text)
> > > > - I think the figures show more than a table does so I like having the results as figures. That said, I can include tables in the supplement. Maybe if I reduce the number of rows of abilities it might help
> > > > - the bolded methods are ones from our paper, we'll point that out
> > > > - Calibration: Figures 1-3 are exactly testing calibration. Figure 1 is a whole interval (or rather whole distribution) calibration test in the form of the KL divergence. I think that's better than looking at the point estimates for the intervals or for the parameter.

---

### Review · Reviewer_NyVn · 2025-06-20

**Summary Of Contributions:**

This paper falls into the category of what I would call applied statistics. The application is modeling of human "abilities"; my understanding of the model is that there are a finite number of *items* (how many? $I$?), and what we can measure is the *response* to these items ($K$ possible responses for all items?), presumably something like answers (=response) to questions (=items) on a standardized test. The *ability* is encoded as a parameter $\\theta$, and the data-driven procedure is that of choosing $\\theta$ such that the resulting $\\theta$-conditional response model "fits the data", so to speak; in a Bayesian setup the posterior distribution of $\\theta$ encodes ability. This is "item response theory" (IRT), as I understand it.

The other key bit of background is computer adaptive testing (CAT), where items are presented sequentially; each step consists of presenting an item and receiving a response, based upon which a running candidate for the ability parameter (or its conditional distribution) is updated. One important design consideration is how to choose the next item to be presented. The authors in section 1 highlight how a standard approach is that of choosing an item to maximize the "local information" (i.e., at step $t$, given ability estimate $\\hat{\\theta}\_{t}$, in step $t+1$, choose item $i$ such that $J\_{i}(\\hat{\\theta}\_{t})$ as given in equation 1 is maximized).

The approach taken by the authors, described as "Bayesian information theoretic", looks to be characterized by equation (5), noting that $\\pi(\\boldsymbol{\\theta}\\vert\\mathbf{x})$ is the "full bank" posterior (aka "true estimate"), i.e., $\\mathbf{x}$ contains all responses to all items; this is of course unknown until all items have been presented, but the idea is to get the running estimate to align with this distribution, where misalignment is measured here by KL divergence. The authors' approach involves estimating the ability distribution as described in section 2.3, and using these estimates to determine a distribution for random sampling of items, as described in equation (11). This looks to be the key element of their proposal, as an alternative to more traditional "greedy" methods for item selection which suffer from poor "item exposure". The authors compare their approach with several more traditional methods on simulated data, using a variety of evaluation metrics, and they report a good balance of exposure and efficiency.

**Audience:**

No

**Broader Impact Concerns:**

Not applicable.

**Claims And Evidence:**

Yes

**Requested Changes:**

Please see the comments raised above. Aside from the minor revisions, I don't have any major requests, aside from considering whether or not TMLR is actually an appropriate venue for this paper.

**Strengths And Weaknesses:**

Having described my understanding of this paper in the preceding field, my thoughts on the strengths and weaknesses of this paper are as follows.

For the most part the paper is clearly written; I get what they are trying to accomplish, and I understand what their proposed method is, and the motivations behind it. The claims, I feel, are for the most part well-supported by the evidence in the paper, though I do not know if people working in the field of CAT/IRT would be convinced by the simulation-based experiments here.

The main concern I have is whether this paper will have an audience with TMLR. I think the "solid claims, clear writing" criterion is mostly satisfied, but I wonder if the "some of the TMLR readership would be interested in this paper" criterion is satisfied by this paper. TMLR is a medium for machine learning research; while one can say that a subset of machine learning techniques are used in the application considered here, this paper is truly just an application of standard techniques to a very specific real-world problem setting. I seriously wonder if TMLR readers would be interested in the results presented here; it really seems like more application-oriented journals would be a better fit (e.g., Behaviormetrika, Behavior Research Methods, something along these lines?).

The paper represents a genuine effort, but it seems like TMLR might not be the proper fit. In any case, below I provide a few technical notation points I tripped up on, which I think merit revision.

- The left-hand side of equation (1) should be $J\_{i}(\\hat{\\theta}\_{t})$, without the $\\theta = \\hat{\\theta}\_{t}$ subscript; having $\\theta$ on the left-hand side clashes with $\\theta$ on the right-hand side.

- On page 1, $K$ is not explicitly defined.

- On page 1, $I$ appears (the overall Fisher information), but this clashes with $I$ on page 3 (equation (3)), namely the number of items. In addition, the $I$ on page 3 is not really defined explicitly; I think the number of items merits a direct definition and its own symbol.

- The symbol $P(I, t)$ on page 3 seems kind of meaningless; it is not defined explicitly, and really the text sufficiently describes what the alpha vector is.

- Last paragraph of section 2.2: $\\{x\_{\\alpha\_{t+1}}\\}\_{s=t+1}^{I}$. Something is wrong with the indexing here.

---

> ### Author Response · Authors · 2025-06-25
> **Response to NyVn**
>
> Thanks for the detailed review and for catching some typos which we have resolved in the revision that we're working on. Your summary of IRT is good. There are some nuances relating to having incomplete data. Actually CAT itself is an application of IRT models to incomplete data. This incomplete data problem is the crux of what we are looking at in the manuscript.
>
> In "calibrating" the initial IRT model to a given sample of respondents, one models the response patterns for the sample of populations (aiming to extrapolate the model to the population at large given certain usual sampling assumptions). Fitting the initial model defines a latent continuous scale that provides a rank transformation for the collection of respondents. When taking that fitted model and using it in order to *score* new respondents, as in CAT, one is placing the new respondent somewhere in the scale that is defined using the initial calibration sample.
>
> In both cases there is a missing value problem.  Our manuscript is focused on the CAT/scoring application of IRT where we provide a solution to the missing value problem (Eq 7 rather than Eq 4 is the right expression for the ability) in terms of an EM algorithm. The true ability estimate (that is consistent with the scale defined at the initial IRT model calibration) is the score that you get if the respondent answers all items. The EM algorithm computes the expectation of this quantity (Eq 7). Then our method makes the analogy to model selection based on discrepancy against the *true model* which in this case is the true ability estimate of Eq 7.  Which takes me into replying to your general point about relevance to TMLR readers. Here is a recent NeurIPS paper (that I wasn't aware of at the time of original submission) that also acknowledges that Eq 7 is the correct true estimate:
>
> * Zhuang, Y., Liu, Q., Zhao, G., Huang, Z., Huang, W., Pardos, Z., … Li, X. (2023). A Bounded Ability Estimation for Computerized Adaptive Testing. Thirty-Seventh Conference on Neural Information Processing Systems. Retrieved from https://openreview.net/forum?id=tAwjG5bM7H
>
> Their methodology is a greedy approach motivated by optimizing a different criterion (the point L2 error), using likelihood gradient information, that bears some similarity to BOED methods. We'll comment on this method in the new revision.

---

> > ### Author Response · Authors · 2025-06-25
> > **Response to NyVn part 2**
> >
> > CAT and its natural sibling IRT may not be hot topics in the ML world right now but they are the silent engines behind a lot of real-world applications. For example the FDA/NIH have registered guidelines (called PROMIS) for designing IRT models and administering them using CAT or short form item subsets.  For this reason alone I think that the topics of CAT and IRT are applicable to the readers of TMLR.
> >
> > Additionally, here is a list of recent IRT/CAT manuscripts that appeared recently in big ML conferences, almost entirely accepted into the main conference:
> >
> > * Susanne Frick, Amer Krivosija, Alexander Munteanu Proceedings of The 27th International Conference on Artificial Intelligence and Statistics, PMLR 238:1234-1242, 2024.
> > * Joshua C. Chang, Carson C. Chow, Julia Porcino Proceedings of The 26th International Conference on Artificial Intelligence and Statistics, PMLR 206:3961-3976, 2023.
> > * Nguyen, D., & Zhang, A. Y. (2022). A Spectral Approach to Item Response Theory. In A. H. Oh, A. Agarwal, D. Belgrave, & K. Cho (Eds.), Advances in Neural Information Processing Systems. Retrieved from https://openreview.net/forum?id=1ItkxrZP0rg
> > * James Sharpnack, Kevin Hao, Phoebe Mulcaire, Klinton Bicknell, Geoff LaFlair, Kevin Yancey, Alina A. von Davier Proceedings of Large Foundation Models for Educational Assessment, PMLR 264:121-135, 2025.
> > * Tio, S., & Varakantham, P. (2023). Training Reinforcement Learning Agents and Humans with Difficulty-Conditioned Generators. Second Agent Learning in Open-Endedness Workshop. Retrieved from https://openreview.net/forum?id=UBxh6uhyuc
> > * Liu, Z., Zhuang, Y., Liu, Q., Li, J., Zhang, Y., Huang, Z., … Wang, S. (2024). Computerized Adaptive Testing via Collaborative Ranking. The Thirty-Eighth Annual Conference on Neural Information Processing Systems. Retrieved from https://openreview.net/forum?id=5Fl4zgXbsW
> > * Zhuang, Y., Liu, Q., Zhao, G., Huang, Z., Huang, W., Pardos, Z., … Li, X. (2023). A Bounded Ability Estimation for Computerized Adaptive Testing. Thirty-Seventh Conference on Neural Information Processing Systems. Retrieved from https://openreview.net/forum?id=tAwjG5bM7H
> > * Li, S. (2025). Trustworthy AI Meets Educational Assessment: Challenges and Opportunities. Proceedings of the AAAI Conference on Artificial Intelligence, 39(27), 28637-28642. https://doi.org/10.1609/aaai.v39i27.35089
> > * Zhuang, Y., Liu, Q., Huang, Z., Li, Z., Shen, S., & Ma, H. (2022). Fully Adaptive Framework: Neural Computerized Adaptive Testing for Online Education. Proceedings of the AAAI Conference on Artificial Intelligence, 36(4), 4734-4742. https://doi.org/10.1609/aaai.v36i4.20399
> > * Wang, C., Yang, S., Song, S., Wang, Z., Ma, H., Zhang, X., & Jin, B. (2025). Explicit and Implicit Examinee-Question Relation Exploiting for Efficient Computerized Adaptive Testing. Proceedings of the AAAI Conference on Artificial Intelligence, 39(12), 12685-12693. https://doi.org/10.1609/aaai.v39i12.33383
> > * Yu, J., Zhuang, Y., Huang, Z., Liu, Q., Li, X., Li, R. &amp; Chen, E.. (2024). A Unified Adaptive Testing System Enabled by Hierarchical Structure Search. <i>Proceedings of the 41st International Conference on Machine Learning</i>, in <i>Proceedings of Machine Learning Research</i> 235:57803-57817 Available from https://proceedings.mlr.press/v235/yu24r.html.
> > * Yang Liu, Alan Medlar, and Dorota Glowacka. 2023. What We Evaluate When We Evaluate Recommender Systems: Understanding Recommender Systems’ Performance using Item Response Theory. In Proceedings of the 17th ACM Conference on Recommender Systems (RecSys '23). Association for Computing Machinery, New York, NY, USA, 658–670. https://doi.org/10.1145/3604915.3608809
> >
> > One of these papers is on a different set of ML techniques that was inspired by IRT models.  I have recently reviewed manuscripts that used IRT in order to holistically compare the performance of different models. CAT also has relationship to BOED which another reviewer noted - there is room for cross-pollination between to two groups of researchers. In the updated manuscript, I'll comment on BOED and the closely related field of active learning which some of the listed a manuscripts above mention.
> >
> > Ultimately relevance is up to you, the rest of the reviewers, and the editor. However, based on the existence of related papers in the mainline ML conferences and mathematical similarities between IRT/CAT and other machine learning methods, I think that our manuscript is relevant to TMLR readers.

---

> > > ### Comment · Reviewer_NyVn · 2025-07-07
> > > **Re: Response to NyVn**
> > >
> > > I thank the authors for their response and the additional references, which I have found quite informative for making my final recommendation.

---

### Author Response · Authors · 2025-05-16
**Cover Letter**

While computer adaptive testing (CAT) and item response theory (IRT) are not currently hot topics in the machine learning community, their daily impact is huge because of the ubiquity of their real life utilization. I want to note that CAT and IRT have been the subject of several manuscripts over the last three years at venues such at NeurIPS and AISTATS. For this reason we believe that the subject of this manuscript is relevant to readers of TMLR.

---

### Author Response · Authors · 2025-06-30
**New Introduction**

Apologies as I originally promised a revision this weekend. It will take me a couple more days to put everything back together. For now, I am posting the new Introduction which will address some of the concerns raised by xpBm and ifxV. In broad strokes I worked on making the introduction itself less technical and more readable. The technical content that was in the old introduction is now in a separate section called "Preliminaries." Overall the new paper has organization that is more similar to the bulk of ML conference papers.

See below:

## Introduction
The combination of Item Response Theory (IRT) and Computer Adaptive Testing (CAT) forms the domi-
nant methodology for modern assessment administration and analysis. High profile examples of this pairing
include the Graduate Management Admission Test (GMAT) (Kingston et al., 1985; Rudner, 2010), the
nursing National Council Licensure Examination (NCLEX) (Woo & Dragan, 2012), the National Registry
of Emergency Medical Technicians (NREMT) (Ventura et al., 2021), and the Armed Services Vocational
Aptitude Battery (ASVAB) (Segall & Moreno, 1999). IRT/CAT also features in many healthcare con-
texts because of its adaptation in Patient Reported Outcomes Measurement Information System (PROMIS)
instruments (Cella et al., 2010; Segawa et al., 2020) that are widely used in FDA-regulated trials.

### Item Response Theory (IRT)
IRT, a generative latent-variable modeling framework, models how a respondent of a given ability level
might respond to to each item in a testing bank. In IRT, an ability (canonically denoted θ) is a theoretically
continuous valued parameter. The initial step for developing an IRT model involves creating a large pool of
items that are topically grounded in a construct being measured. These items are then administered to a
sizable and diverse sample of respondents, producing a dataset of item responses for model calibration. In the
process of fitting an IRT model to the set of item responses, each item’s specific parameters are determined.
Self-consistently, the ability for each of the respondents is also determined. Due to this coupling, the ability
statistics for the calibration sample encode into the item-specific parameters.

Fundamentally, IRT maps each respondent’s set of discrete item responses to a lower (usually single) dimen-
sional latent space. In this manner, IRT models are nonlinear factorization models, related to probabilistic
autoencoders (Chang et al., 2019; 2023). The latent space places each calibration sample respondent into a
relative population rank. The goal of IRT is to apply such pre-trained models to new respondents, ranking them relative to the
respondents used in model calibration. To do so, item parameters from calibration are held fixed and new
responses for a given respondent are scored by solving an associated inverse problem for the ability parameter.
The possibly large item bank developed for the IRT model ideally has content coverage throughout the entire
range of possible abilities. Administering a large item bank is burdensome for all parties involved. In the
vicinity of any fixed ability parameter, however, the number of items is relatively small. CAT exploits this
fact.

---

### Author Response · Authors · 2025-06-30
**New intro 2**

### 1.2 Computer Adaptive Testing (CAT)
The goal of computer adaptive testing (CAT) is to efficiently and accurately estimate a respondent’s ability
by using only the most relevant questions from a possibly large item battery. This selection is performed
sequentially based on a running estimate of the respondent’s ability. Selection methods mainly differ on
the specific statistical objective being optimized. Generally, individual items are judged based on some
measure of the degree to which they may improve the fidelity of the respondent’s ability estimate. Most
commonly, items are chosen greedily – while efficient, this type of selection procedure has the pitfall of poor
item exposure.
Item exposure refers to the rate at which individual items in a testing bank are presented across multiple
administrations. When exposure is poor, the effective instrument administered by the CAT is a limited
subset of the items in the original bank. In unison with commonly-used improper scoring rules, this condition
biases the resulting ability estimates. Having a small number of effective items also implies stereotypical
item trajectories, making such instruments easier to game.

CAT methodologies select items based on a running estimate of a test-taker’s ability. However, this estimate
is unreliable at the beginning of the test, which in turn makes the statistical measures used to compare items
noisy. For this reason, simply choosing the item that appears statistically best (a "greedy" approach) may
not be ideal. A more effective strategy may be to hedge, selecting items that are useful across a wider range
of potential ability levels.

In this manuscript we provide a methodology for hedging that is based on viewing item selection as a model
selection problem. Each item implies a different model for the respondent’s ability at the next step of the
test. As a consequence of viewing the problem through these lens, we both motivate a new item selection
criterion based on the information theoretic model discrepancy for the ability estimate, and a stochastic selection procedure
where the Frequentist statistics of item probabilities correspond to Bayesian model averaging statistics of
the item-wise implied ability estimates.

### 1.3 Work Disability Functional Assessment Battery (WD-FAB)
As concrete tests of our methodology we used the eight independent IRT models, and their associated item
banks, present in the WD-FAB (Meterko et al., 2015; Marfeo et al., 2016; 2019; Chang et al., 2022; Marfeo
et al., 2018; Jette et al., 2019; Porcino et al., 2018). The WD-FAB characterizes whole body and mental
function across four physical instruments and four mental instruments. The item banks consist of questions
that ask about a range of everyday activities, such as emptying a dishwasher, walking a block, turning a
door knob, speaking to someone on the phone, and managing under stress. Accepted responses were graded
on either four or five option ordinal Likert scales.

The intended use of this instrument is to provide standardized and reliable information about an individual’s
functional abilities to help inform SSA’s disability adjudication process. The WD-FAB provides eight scores
across two domains of physical and mental function that are relevant to a person’s ability to work.
As an application where item exposure is important, the eight independent models that comprise the WD-
FAB are an ideal testing ground for our methodology.

---

> ### Comment · Reviewer_xBpm · 2025-06-30
>
> I think TMLR allows to revise the submission and upload the revised version. The authors should upload the revised version in the submission portal, highlighting the revised paragraphs.

---

> > ### Author Response · Authors · 2025-06-30
> > **Re:**
> >
> > Thanks, I will post the revised version I'm a day or two, I need more time than I previously thought.

---

### Author Response · Authors · 2025-07-07
**General description of manuscript revisions**

Reiterating  and expanding on what I posted in the relevant metadata field when I first posted the revision last week:

This revision is an extensive rewrite, shifting the structure of the manuscript to better mimic that found in usual ML conference papers. The Introduction section in particular is rewritten for clarity, in a less technical manner.

The manuscript now has more consistent emphasis on the focal novelty of the method: stochastic selection where the Frequentist item statistics yield Bayesian model averaging in ability estimate space. The exposition before was backwards before in a way - it's the stochastic motivation that led to me defining the criterion the way that I did rather than the criterion yielding a stochastic selector.

Prior art has its own dedicated subsection now where I put the Fisher information stuff in plus introduced the baseline methods for comparison. I also comment on BOED there.

 I decided not to have a dedicated notation section and instead decided `) to change some of the notation so it is more obvious what the symbol means $N_{items}$ versus $I$ for instance 2) Every symbol should now have a description when it first appears. Sorry about the confusion on the first draft.

Additionally I also added some more citations for IRT background.

---

### Decision · Action_Editor_WxEM · 2025-08-02

**Recommendation:** Reject

**Additional Comments:**

I recommend to reject this paper because I judged that this paper requires a major revision.  The biggest problem is in the readability: even after reading the abstract and the introduction (of the revision), what the authors claim as contributions is unclear.  Furthermore, it is hard to assess the significance of the paper.  I quote Reviewer ifxV's final concerns, which must be addressed in a resubmission.

"As noted in my comment below, the results are very hard to judge and read, they also only rely on testing the posterior means and reporting the posterior variance, which is not in line with modern Bayesian workflows. The authors claim that the method is widely applicable to various IRT models and beyond, but they only show one simulation on a small grid of one-dimensional parameters. For the real data, only item exposure results are reported for a single IRT model. Combined with the general lack of clarity in writing, I have doubts that this evaluation is solid even for the IRT/CAT fields."

Although the authors made significant revision, the readability issues persist.  The problem is in that the authors use too many domain-specific words, with which reviewers and AE are not familiar.  In the introduction, IRT and CAT are explained by using other domain-specific words, which is not helpful.  For example, the meanings of the following terms should be clearly explained in the introduction: IRT instrument, respondent, Frequentist item sampling statistics, next-item ability estimates, item bank ability estimate, battery, SSA.  I suppose some are general words but the terminology is very different from what ML researchers use, so they must be explained with plain wording.  Perhaps giving a very concrete example (who evaluates whose abilities for what purpose under what situations.  In such an example, what instrument, item, bank, battery, etc. correspond to) would be helpful.  Since the reviewers judged that the contribution is an application of existing ML methods to a specific application domain, the target applications must be fully understood by typical ML readers.

Also, there are confusing sentences everywhere.  I give two in the abstract, but please check the other parts and fix logically incorrect sentences.

1. "Computer Adaptive Testing (CAT) aims to accurately estimate an individual’s ability using
only a subset of an Item Response Theory (IRT) instrument. A secondary goal is to ensure
diverse item exposure across dierent testing sessions, preventing any single item from being
over or underutilized. "

I'd suppose that CAT aims to accuracy estimate individual's ability, and there's no secondary goal.   Diverse item exposure is necessary for accuracy (i.e., for making the estimate unbiased), so it's not secondary goal but a subgoal.  If diverse exposure is really a secondary goal even after CAT achieves optimal accuracy, the authors should explain for what this secondary goal is required.

2. "the KL divergence between the unknown next-item ability estimate and the unknown true full item
bank ability estimate."

KL divergence is defined not for two estimates but for two distributions.  In the field of CAT/IRT, is a posterior distribution called an estimate?  Even if so, it's better to adjust to the ML terminology when publishing a paper in the ML domain.

Since all reviewers say that the paper has audience, I would encourage the authors for resubmission after a major revision.

**Audience:**

Yes

**Audience Explanation:**

All reviewers acknowledged.

**Claims And Evidence:**

No

**Claims Explanation:**

As pointed out by Reviewer ifxV, it is hard to judge due to the readability issue.

**Resubmission Of Major Revision:**

The authors may consider submitting a major revision at a later time.

---

> ### Author Response · Authors · 2025-08-03
> **Re:**
>
> Thank you for your guidance, we will work on resubmitting after making another set of revisions.
>
> Re: estimate, by estimate we refer to the whole distribution as an estimate, which is an estimate in the Bayesian sense of the word.